# Uncertainty Quantification in Water Resource Systems Modeling: Case Studies from India

**Shaik Rehana** [1] , **Chandra Rupa Rajulapati** [2] , **Subimal Ghosh** [3] , **Subhankar Karmakar** [4] **and Pradeep Mujumdar** [5,*]

[1]   Lab for Spatial Informatics, International Institute of Information Technology Hyderabad, Hyderabad 500032, India; rehana.s@iiit.ac.in
[2]   Centre for Hydrology, University of Saskatchewan, Saskatoon, SK S7N1K2, Canada; chandra.rajulapati@usask.ca
[3]   Department of Civil Engineering, Indian Institute of Technology Bombay, Mumbai 400076, India; subimal@civil.iitb.ac.in
[4]   Environmental Science and Engineering Department, Indian Institute of Technology Bombay, Mumbai 400076, India; skarmakar@iitb.ac.in
[5]   Department of Civil Engineering, Interdisciplinary Centre for Water Research, Indian Institute of Science Bangalore 520011, India
*   Correspondence: pradeep@iisc.ac.in; Tel.: +91-80-2360-0290

**Abstract:** Regional water resource modelling is important for evaluating system performance by analyzing the reliability, resilience and vulnerability criteria of the system. In water resource systems modelling, several uncertainties abound, including data inadequacy and errors, modeling inaccuracy, lack of knowledge, imprecision, inexactness, randomness of natural phenomena, and operational variability, in addition to challenges such as growing population, increasing water demands, diminishing water sources and climate change. Recent advances in modelling techniques along with high computational capabilities have facilitated rapid progress in this area. In India, several studies have been carried out to understand and quantify uncertainties in various basins, enumerate large temporal and regional mismatches between water availability and demands, and project likely changes due to warming. A comprehensive review of uncertainties in water resource modelling from an Indian perspective is yet to be done. In this work, we aim to appraise the quantification of uncertainties in systems modelling in India and discuss various water resource management and operation models. Basic formulation of models for probabilistic, fuzzy and grey/inexact simulation, optimization, and multi-objective analyses to water resource design, planning and operations are presented. We further discuss challenges in modelling uncertainties, missing links in integrated systems approach, along with directions for future.

**Keywords:** reservoir operation; stochastic dynamic programming; fuzzy optimization; reservoir-river system; water quantity-quality management; climate change

## 1. Introduction

Water resource management is about the integration of various disciplines of hydrology for the planning, management and optimum utilization of water resources following the competing needs and demands of society. Integrated water resource management consists of four dimensions: (i) natural element of water resources, considering the entire hydrological cycle and various components of it such as rainfall, water in rivers, etc.; (ii) water users and stakeholders, including socioeconomic interests; (iii) variability of water resources and users, such as spatial mismatch of water availability between upstream and downstream river plains; (iv) temporal variability of water availability and demands [1].

Globally regional hydrologic systems have long struggled for many decades with the planning and management of water resources under growing population, increasing demands and climate change [2]. River water resource systems are under great stress as a result of unsustainable consumption patterns and poor management practices [3]. The need for a regional water resource management model accounting for water availability and demands, water quantity and quality has become prominent in recent years under climate signals [4]. Based on several scientific studies, climate change is likely to affect various subsystems of regional water resource systems, such as water availability for consumer needs and food production, irrigation water demands, hydropower, water quality, etc., under an increase in temperatures and changes in precipitation patterns [5]. Climate change has been identified as one of the major driving forces in regional water resource systems management by several studies globally [6–8] and in India [3,9,10].

A regional water resource management model is an integration of a water quantity and quality estimation model, a water demand estimation model along with a decision making model [11]. For instance, a hydrological model is used to estimate the water availabilities in terms of inflows; demand estimation models estimate factors such as drinking, irrigation and hydropower; water quantity and reservoir operation models are used to estimate the optimal release policy and water allocations of reservoir users; and water quality management models are used to estimate the optimal treatment policies (Figure 1). Figure 1 shows a single reservoir–river system accounting for the upstream catchment flows, evapotranspiration, overland flows, infiltration and in the downstream side water withdrawals and return flows from irrigation. An integrated operation of reservoirs of a complete river system, starting from the furthest upstream reservoir to the furthest downstream reservoir, should include inflows to each reservoir, evaporation losses, power draft, releases, withdrawals and overflows in the reservoir operation. While simulating and integrating the reservoir operation of major river systems, the inflow to any particular reservoir should account for uncontrolled intermediate catchment flows, irrigation return flows and controlled flows from the upstream reservoir [12].

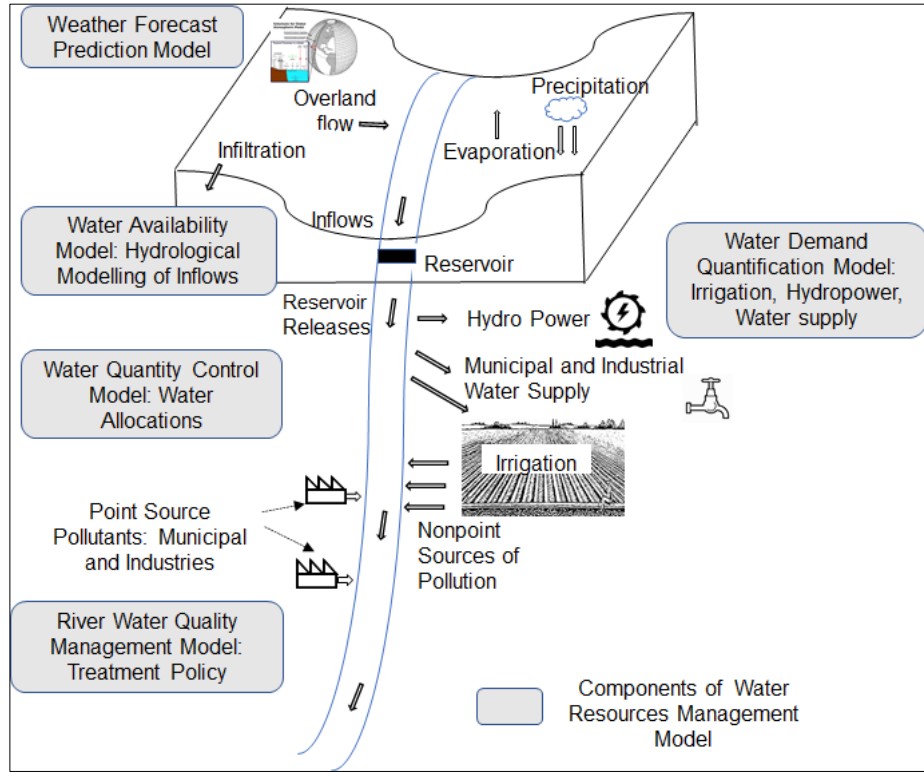

**Figure 1.** An integrated regional water resource system management model.

Integrated regional water resource management models have evolved to secure water resource systems at the basin scale in terms of water quantity and quality, accounting for water availability and the demands of various users [1]. In this context, water resource systems models have been evolved in the past four decades in several aspects of single and multi-purpose reservoirs, optimization models, knowledge-based decisions, real time operations, imprecision and uncertainty quantifications and climate change [13]. Several review papers have articulated the evolution of water resource management systems modeling, focusing on several key aspects in terms of optimization models, and concluded the research gaps between the developed models [11,14,15]. One of the pioneering review papers on reservoir systems analysis was by Simonovic [16], in which the gaps between research studies and application of systems approach in practice were discussed and an optimization model for reservoir sizing and the inclusion of knowledge-based technologies in single-multipurpose reservoir analysis was recommended. Furthermore, most of the earlier review papers articulated on the evolution of water resource management modeling at basin scale [17] integrated water resource optimization models [18,19]. Very recent review studies focused on the application of evolutionary algorithms and metaheuristic optimizations for optimal strategies of the planning and management of water resource systems [20–24]. In this context, Mohammad-Azari et al. [23] have reviewed the application of Genetic Programming to solve water resource systems analysis and stressed on the capability and superiority of evolutionary algorithms in solving reservoir operation problems. Few earlier review papers focused on reservoir operation challenges related to inflows [25], simulation and optimization techniques [26].

The integrated regional water resource management models are associated with various forms of uncertainties accumulating from various stages of decision making [27]. Uncertainties arise at each stage of the modelling and decision-making process due to random nature of input variables, various parameters and models, imprecise goals of the users, priorities and social importance in decision making by various stakeholders. Addressing these uncertainties is very important for precise decision making and to avoid the failure of water resource system management [16]. The inclusion of uncertainties of reservoir inflows in the water resource systems models was one of the basic studies and have implemented by several researchers by considering inflow as stochastic variable [28]. The next prevailing uncertainty in reservoir operation is imprecise goals of the users, which has been conventionally addressed using fuzzy set theory [29]. Identifying and addressing various sources of uncertainties is one of the crucial tasks in water resource modelling to have better operating policies with more dependability and flexibility in decision making. Review papers which can articulate various studies of water resource management and associated uncertainties are limited in the literature. Ahmad [15] reviewed reservoir operation models with fuzzy optimization along with other optimization methods such as Artificial Neural Network (ANN), Genetic Algorithm (GA), artificial bee colony and Gravitational Search Algorithm (GSA). A comprehensive review which can include the uncertainty quantification in water resource systems modeling, various approaches so far applied, research gaps and challenges is lacking in the literature. In this article, we review water resource management systems models to address various sources of uncertainties by highlighting key findings and identify important future research directions which can improve the understanding of water resource planning and management.

India has large regional mismatches between water availability and demands, with increasing withdrawals from surface and subsurface sources rising to unsustainable conditions [30]. India is an agriculture-dominated country and about 70% of the population's employment and economy depends on agriculture sector. The timely supply of irrigation water with sufficient quantity is challenging given the spatial and temporal mismatches of river water availabilities, increasing drinking and industrial water demands under population growth and pressure to increase crop yields. The determination of optimal water allocations for various sectors to fulfill various demands is of primary interest for most of the reservoirs of India. Tremendous population growth, rapid urbanization, alterations in agricultural patterns, unplanned growth of industries and failure of maintaining the environmental standards are the major causes for poor river water quality systems in India [31]. The present research

article explores some of the Indian case studies carried out in the field of water resource management and uncertainty quantification. Sources and approaches to address uncertainties in the context of water resource management and modelling are discussed with a focus on Indian case studies. Furthermore, missing links in modelling, challenges remaining and future directions are noted.

## 2. Reservoir Operation and Associated Uncertainties

Reservoir operation has gained attention in water resource engineering for more than four decades [32]. The reservoir operation systems models vary according to various components of consideration such as drinking water supply, irrigation, hydropower, low-flow augmentation, aquaculture, navigation along with flood control and management. Fundamentally, a regional water resource systems management model is an integration of a reservoir operation model to define the possible releases following the storage continuity equation and an optimization model to define the optimal water allocation policies, accounting for the conflicting goals of the reservoir users and possible demands (Figure 2).

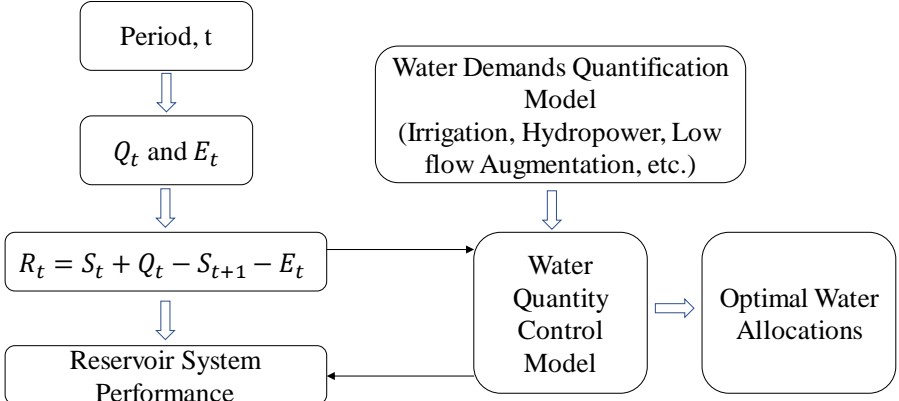

**Figure 2.** Water quantity control model of a reservoir.

For a given period of time, t, the inflows to the reservoir ($Q_t$) and evaporation loss from the reservoir during the period, $t$ ($E_t$), storage at the beginning of the period ($S_t$), storage at the end of the period ($S_{t+1}$), the continuity equation forms the basis for the determination of the possible releases (Figure 2). The release during the period, t, $R_t$, is the decision variable, with storage at the beginning of the period, t, $S_t$, as the state variable in the reservoir optimization model with objective as to maximize the total net benefit, $B_t(S_t, R_t)$, during a year T [33]:

$$\text{Maximize } \sum_{t=1}^{T} B_t(S_t, R_t) \tag{1}$$

$$0 \le R_t \le S_t + Q_t \tag{2}$$

$$S_t + Q_t - R_t \le K \tag{3}$$

Equations (2) and (3) represent constraints over the possible release, $R_T$, restricting it to the total water available in storage in period $t$ (Equation (2)), and the end of period storage ($S_{t+1}$) is restricted to the live storage capacity, (K) (Equation (3)). In general, the optimization model has to be solved recursively until it yields a steady state policy within a few annual cycles [3]. In this single objective reservoir operation model, the most influential variable for optimal release is the reservoir inflows (water available for release) and it is highly uncertain, due to the upstream catchment rainfall uncertainty and other basin characteristics. In addition, other hydrological variables such as evapotranspiration, soil moisture, ground water flows, etc. which define crop water demands in the downstream command area are also burdened with uncertainty due to randomness which can cause stochastic or aleatory uncertainty in the reservoir operation [28].

In this context, various studies considered the input variables of a reservoir operation model as having a random nature and explicitly included in the optimization model through their probability distributions [34]. The hydrologic variable uncertainty due to randomness has been addressed by various authors by considering the reservoir inflow to follow a one-step Markov process through transition probabilities over Indian case studies [35–37]. Conventionally, the uncertainty due to the randomness of inflows in reservoir operation is addressed by applying stochastic dynamic programming (SDP) [32]. In one of the pioneering works by Vedula and Mujumdar [38], a reservoir operation model based on SDP was developed to find the optimal water allocations for irrigation under multiple crops scenarios, where reservoir storage, inflows, and soil moisture are treated as state variables in the decision-making process for Malaprabha reservoir, Krishna basin, Karnataka state, India. Ravikumar and Venugopal [39] developed an optimal operation model using simulation and SDP combination, where both demand and inflow are considered as stochastic and both are assumed to follow first order Markov chain model, which is demonstrated with the Periyar Vaigai irrigation system as one of the typical south Indian irrigation systems of India. Mujumdar and Kumar [12] developed an integrated reservoir operation tool for providing the operation of the eight major reservoirs of Narmanda river basin, India. The study developed a simulation model with eight major reservoirs, viz., Matiyari, Bargi, Barna, Tawa, Indira Sagar, Omkareshwar, Maheshwar and Sardar Sarvovar, as shown in Figure 3. A computer simulation model was developed starting from the furthest upstream reservoir (Matiyari) to the furthest downstream reservoir (Sardar Sarovar) by accounting for inflows to the reservoir (including uncontrolled intermediate catchment flows, irrigation return flows and controlled flows from upstream reservoirs), evaporation losses, power draft, releases, withdrawals and overflows during every period until the end of simulation.

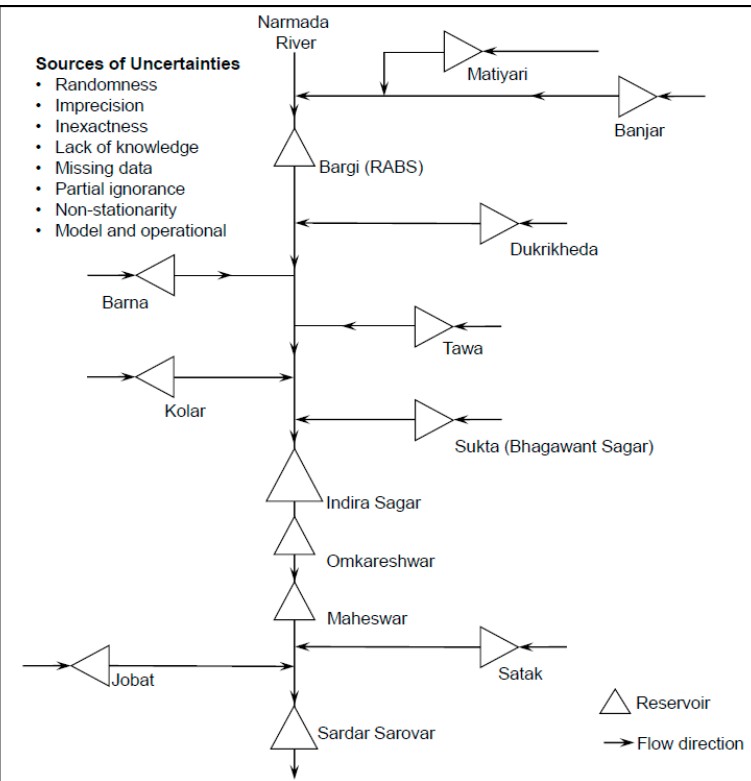

**Figure 3.** Integrated reservoir operation with major and medium reservoirs in Narmada river basin, India, modified from Mujumdar and Nagesh Kumar [12].

Reservoir operation based on SDP has emerged as a promising tool to address the uncertainty of reservoir input variables for various case studies of India, namely Hirakud reservoir [3,40], Malabrapha

reservoir [38], Bhadra reservoir [4,37], Ukai reservoir [41], Kodaiyar Basin [42]. In recent years, various forms of SDP, such as folded dynamic programming [34], two-phase stochastic dynamic programming [43], deterministic dynamic programming (DDP) [44], etc., have become popular for reservoir operation and management in India based on the abilities to improve the net benefit and to overcome the dimensionality issues of SDP.

The reservoir operation stakeholders are often uncomfortable with sophisticated optimization techniques, and need flexibility in specifying the goals and decision making, which causes uncertainty due to imprecision in water resource systems models [45]. Fuzzy logic was identified as an appropriate tool to address the uncertainty due to imprecision in defining the goals of the stakeholders [29]. In this context, fuzzy water allocation models to address the uncertainty due to imprecision in defining the goals of the reservoir users has been widely used all over the world [46] and in India [47–49]. A typical fuzzy optimization model for reservoir operation works on specifying the goals of the users as fuzzy membership functions and the mathematical formulation of a typical reservoir operation as a water quantity control model, following Rehana and Mujumdar [4], which can be expressed as follows:

$$\text{Maximize } \lambda \tag{4}$$

Subject to

$$f(q_\alpha) \geq \lambda \tag{5}$$

$$f\left(q_\beta\right) \geq \lambda \tag{6}$$

$$f(q_\chi) \geq \lambda \tag{7}$$

$$q_\alpha^{Min} \leq q_\alpha \leq q_\alpha^{D} \tag{8}$$

$$q_\beta^{Min} \leq q_\beta \leq q_\beta^{D} \tag{9}$$

$$q_\chi^{Min} \leq q_\chi \leq q_\chi^{D} \tag{10}$$

$$q_\alpha + q_\beta + q_\chi \leq W_A \tag{11}$$

$$0 \leq \lambda \leq 1 \tag{12}$$

where $W_A$ is the amount of water available for allocation, which is the reservoir release, Rt, for a given time period, *t*, from the reservoir operation model (Equations (1)–(3)) (Figure 2). The solution of the resulting optimization problem will be $q^*$ and $\lambda^*$ where $q^* = \{q_\alpha^*, q_\beta^*, q_\chi^*\}$ corresponds to the optimum water allocation among the water users; viz., irrigation ($\alpha$), water quality ($\beta$) and hydropower ($\chi$), and $\lambda^*$ is the maximized minimum satisfaction level in the system. The imprecise goals of reservoir users will be represented using membership functions such as $f(q_\alpha), f\left(q_\beta\right)$ and $f(q_\chi)$ for irrigation, water quality and hydropower, respectively. For each reservoir user, the minimum ($q_\alpha^D, q_\beta^D$ and $q_\chi^D$) and desirable ($q_\alpha^D, q_\beta^D$ and $q_\chi^D$) limits will be specified. The fuzzy optimization model works with the objective function so as to maximize the minimum satisfaction level (Equation (4)) including the imprecise goals of the reservoir users as fuzzy membership functions (Equations (5)–(7)), constraints over water allocations to be within minimum and maximum limits (Equations (8)–(10)), constraints over the total water available for allocation (Equation (11)) and constraints over the satisfaction level ranging between 0 and 1 (Equation (12)).

In this context, fuzzy rule-based reservoir operation models have gained interest to address the uncertainty due to impression in specifying the goals of various reservoir users [50]. Many researchers adopted fuzzy optimization models for optimum water quantity allocations in reservoir operation [45,48,49,51]. A fuzzy rule-based model was developed by Panigrahi and Mujumdar [45] for Malaprabha irrigation reservoir in Karnataka, a single purpose reservoir, where the fuzzy membership functions have been constructed for inflow, storage, demand and release. The inclusion of fuzzy membership in reservoir operation can address the uncertainty due to

imprecision but not the uncertainty due to the randomness of input variables. Therefore, the integration of SDP with fuzzy optimization for optimal reservoir operation has become a promising tool for the development of long-term operating policies in recent years [4,52]. These models are advantageous to address the uncertainty due to the randomness of reservoir inflows by applying SDP and due to the imprecision in specifying the goals of the stakeholders by applying fuzzy optimizations. In this context, a water quantity modelling method was developed by integrating SDP and fuzzy optimization model by Rehana and Mujumdar [4]. This model addresses uncertainty due to randomness and fuzziness combinedly in developing long-term operating policies which has been implemented on Bhadra reservoir, India (Figure 2). Furthermore, such a SDP-fuzzy model was extended by Kumari and Mujumdar [52] for Bhadra reservoir by considering the state variables of reservoir storage and soil moisture as fuzzy variables and reservoir inflow as a random variable in modelling reservoir operation using SDP. By considering the state variables as fuzzy variables in the formulation of SDP, uncertainty due to imprecision originating due to consideration of single representative value of the state variable can be addressed. One of the improvements in the developed model can be the consideration of rainfall and potential evapotranspiration also as stochastic variables along with reservoir inflows, but not as deterministic as considered in the study of Kumari and Mujumdar [52]. In another study by Kumari and Mujumdar [53], a fuzzy set-based performance measure for irrigation reservoir system in terms of fuzzy reliability, fuzzy resilience and fuzzy vulnerability to study the failure/success state of a reservoir system was developed by relating evapotranspiration deficit of the crops and applied on Bhadra reservoir system, Karnataka, India. To this end, the fuzzy-SDP reservoir operation models have advanced in several means in addressing uncertainties of probabilistic and imprecision combinedly. In this context, a few attempts have also been made by adopting fuzzy Markov chain-based SDP models to address the probabilistic and fuzzy uncertainty at the same time by introducing the concept of distribution with fuzzy probability to develop a fuzzy-Markov-chain-based SDP (e.g., [54,55]).

## 3. River Water Quality Management under Uncertainties

A water quality management model is essentially an integration of water quality simulation model and an optimization model to manage the quality of river systems without violating the standards specified by the pollution control agencies. A river water quality control model is necessarily a decision-making process to maintain the ecological stability of the riverine environment involving the pollution control boards (PCBs) and effluent dischargers. In this context, Waste Load Allocation (WLA) models have been evolved for determining the required treatment levels or fractional removal levels for various point and non-point sources of pollutants accounting for the water quality standards specified by PCBs in an economically efficient manner. Majorly, WLA models run with the integration of a river water quality simulation model and an optimization model dealing with the goals of dischargers and pollution control boards [56]. In this context, a surface water quality model is a tool for the better understanding of the mechanisms and interactions between anthropogenic residual inputs and resulting water quality [57]. Water quality simulation models run by accounting for river hydrology and hydraulic variables (streamflow, longitudinal slope, Manning's coefficient, etc.), river water quality parameters (dissolved oxygen (DO), biochemical oxygen demand (BOD), nitrates, temperature, etc.), climate data (air temperature, wind speed, etc.), effluent discharge characteristics (pollutant DO, BOD, temperature, etc.) to simulate the river water quality indicators along the river stretch under consideration [57,58]. Meanwhile, an optimization model considers the resulting water quality for a given pollutant loading along with the goals of the PCBs and industries releasing the effluents [59].

The input variables, such as streamflow, temperature, etc., of water quality simulation models are random variables and therefore are associated with uncertainty due to their randomness [60]. Conventionally, the uncertainty due to randomness in the river water quality variables has been addressed using probabilistic mathematical programming techniques [61]. Another major source of uncertainty is associated with the imprecise goals of the dischargers and PCBs, which is usually

addressed by fuzzy membership functions to represent the satisfaction levels of the users by most of the Indian authors [56,59,62,63].

The pioneering work in the fuzzy river water quality management models was by Sasikumar and Mujumdar [56]. The study developed a Fuzzy Waste Load Allocation Model (FWLAM), addressing the uncertainty due to imprecision in specifying the goals of the dischargers and PCBs. In this context, FWLAMs were evolved to address the uncertainty due to imprecision in specifying the goals of the stakeholders [56] and fuzzy risk minimization waste load allocation model to address uncertainty due to the combined randomness of input variables and fuzziness of decision makers' requirements [64]. The mathematical formulation of a typical river water quality management model can be expressed following Sasikumar and Mujumdar [56] as follows:

$$\text{Maximize } \lambda \tag{13}$$

Subject to

$$f(C_l) \geq \lambda \tag{14}$$

$$f(x_m) \geq \lambda \tag{15}$$

$$C_l^L \leq C_l \leq C_l^D \tag{16}$$

$$x_m^L \leq x_m \leq x_m^D \tag{17}$$

$$0 \leq \lambda \leq 1 \tag{18}$$

where $C_l$ is the concentration level of water quality parameter at check point, $l$; $x_m$ is the fraction removal level for discharger, m; $C_l^L$ and $C_l^D$ are the minimum and maximum permissible levels set by PCBs, respectively; $x_m^L$ and $x_m^D$ are the minimum and maximum possible treatment levels specified by the dischargers, respectively; $\lambda$ as the satisfaction level of PCBs and dischargers. $f(C_l)$ and $f(x_m)$ represent the membership functions of PCBs and dischargers, respectively. The solution of the resulting optimization problem will be $x_m$* and $\lambda$* where $x_m$* corresponds to optimum fraction removal level for each discharger and $\lambda$* is the maximized minimum satisfaction level in the system.

Some improvements in water quality management models were made by Singh et al. [65] by developing an interactive fuzzy multi-objective linear programming model to evaluate optimal treatment efficiencies for various drains located along Yamuna across New Delhi, India. The study allotted weights for DO deficits at each grid point to address the uncertainty in specifying the goals of the decision makers with continuous interaction with decision makers.

Many studies considered risk of low water quality (LWQ) as one of the criteria to represent the goal of the PCBs [59,63,64]. By considering this risk in the river water quality management models, uncertainty due to a combination of randomness in the water quality concentrations along with imprecision in defining the standards was addressed. The risk of LWQ is defined as the probability of a fuzzy event of LWQ [64]. The conventional definition of LWQ is any concentration less than a specified value, say, $c_l^{min}$, the minimum permissible level at check point, $l$. The crisp definition of risk of LWQ, with a water quality indicator as DO, is given as:

$$r_l = P(c_l < c_l^{min}) \tag{19}$$

where $r_l$ is the risk of LWQ at check point, $l$; $c_l$ is the DO level at check point, $l$; $c_l^{min}$ is the minimum permissible level of DO at check point, $l$; $P(c_l < c_l^{min})$ is the probability associated with the occurrence of the LWQ event. The fuzzy risk of LWQ is defined as the probability of occurrence of the fuzzy event of LWQ. Fuzzy risk can be expressed as the expected degree of failure [64].

$$r_{il} = \int_0^\infty \mu_{wil}(c_{il}) \, f(c_{il}) \, dc_{il} \tag{20}$$

where $\mu_{wil}(c_{il})$ is the membership function of the fuzzy set, $W_{il}$ of LWQ and $f(c_{il})$ is the probability density function (PDF) of the concentration level, $c_{il}$, for water quality indicator, $i$, at the checkpoint, $l$ in the river system. Based on the PDF, $f(c_{il})$ of the LWQ indicator, $i$, and the membership function $\mu_{wil}(c_{il})$ of the fuzzy set, $W_{il}$, of LWQ, direct or numerical integration may be performed to evaluate the fuzzy risk, $r_{il}$.

Sasikumar and Mujumdar [64] developed a fuzzy risk approach to address both uncertainty due to randomness and uncertainty due to imprecision of the goals by considering the probability of risk of LWQ as fuzzy event. The study was implemented over the Tunga-Bhadra river stretch, India to estimate optimal fractional removal levels of the dischargers. A fuzzy risk minimization model was solved by Ghosh and Mujumdar [63] to minimize the risk of LWQ using a non-linear optimization model of Probabilistic Global Search Laussane applied to the Tunga-Bhadra river system, India.

In a conventional fuzzy optimization model, the membership parameters are assumed to be fixed and values are assigned based on experience and judgement and are thus highly subjective; for instance, the lower bound of DO is assigned as 5 mg/L and the upper bound is 8 mg/L. In general, such membership parameters are defined based on the minimum and maximum permissible levels of water quality standards, which may vary for each criterion such as public water supply, agricultural and industrial water supplies, etc. [62]. This results in uncertainty in the membership parameters, which can be considered as the next level of fuzziness in the fuzzy optimization models [62]. To address the uncertainty in the membership parameters, Karmakar and Mujumdar [62] developed a grey fuzzy waste load allocation model by considering the membership parameters as interval grey numbers to represent as imprecise membership function. The study was implemented over Tunga-Bhadra river system, India, by considering the imprecise fuzzy membership functions, which provided the optimal treatment policy and satisfaction levels, both in the form of interval numbers, allowing the decision-maker to select various alternatives required in a particular situation. A conventional approach to solve grey optimization models is the two-step sub model method [62,66,67], which bifurcates the parent uncertain model into two daughter models, one for the least favorable case and another for the most favorable case. However, Rosenberg [68] and Yadav et al. [69] found issues such as infeasibility, non-optimality and fat solutions in the two-step method. Any derived problem of an interval/grey model by fixing a deterministic value of available interval numbers is known as the subproblem of the parent model [70] or a deterministic equivalent of the interval/grey model. If the extreme optimum solutions of all such subproblems have significant differences with the solutions obtained from a given technique (two-step method in this case), then the solutions are known as fat solutions [71], which necessarily implies a set of very uncertain outputs.

Huang and Cao [72] further developed a three-step method to resolve the infeasibility of the solution in the two-step method, but made the issue of non-optimality more severe [73]. Yadav et al. [71] proposed an interval-valued integer programming model based on interval analysis to overcome the issues of two-step and three-step methods. Algorithms based on interval analysis are computationally more rigorous than grey analysis, but pave the way for an effective and powerful methodology to quantify the inexact or grey uncertainty. Therefore, interval analysis-based scalable algorithms have the potential to make conventional uncertainty quantification techniques such as probabilistic or fuzzy redundant. In this context, 'Imprecision' is a representation of disjunctive information, which is characterized by a set of possible values for which the actual values are known to exist [74]. This characteristic of ordered disjunctive information has been incorporated in the Fuzzy Set Theory. As per the literature of fuzzy mathematics, the 'imprecision' is analogous to 'vagueness,' [75], which is a linguistic uncertainty and is often represented with fuzzy membership functions. On the other hand, 'Inexactness' is another representation of uncertainty when the exact value is unknown; however, the range within this value exists is known [69]. The concept of inexact uncertainty is relatively new and is extensively used in grey/interval systems. Inexactness may be represented with interval grey numbers, where lower and upper bounds are known, but the distribution information is unknown.

Another source of uncertainty is partial ignorance resulting from missing or inadequate data in a time series of hydrological or water quality variables, which forms the input variables for a water quality simulation model. Rehana and Mujumdar [59] developed an Imprecise Fuzzy Waste Load Allocation Model (IFWLAM) to address the uncertainties not only due to randomness and fuzziness but also due to missing or inadequate data by considering the input variables as interval grey numbers. The developed model was implemented on Tunga-Bhadra river, India. A grey fuzzy risk of LWQ was introduced in the WLAM, which is capable of evaluating grey fuzzy risk with corresponding bounds of DO, rather than specifying a single value of risk. The consideration of fuzzy risk as an interval grey number results in a range of fractional removal levels for the dischargers, which enhances flexibility in decision making (Table 1).

**Table 1.** Results from the IFWLAM optimization models of upper and lower limit of fractional removal levels for various dischargers along Tunga-Bhadra River, India.

| Discharger | Risk Minimization Model (Ghosh and Mujumdar [63]) | Fractional Removal Levels | |
|:---:|:---:|:---:|:---:|
| | | Lower Limit | Upper Limit |
| 1 | 0.77 | 0.69 | 0.69 |
| 2 | 0.77 | 0.68 | 0.69 |
| 3 | 0.65 | 0.35 | 0.68 |
| 4 | 0.77 | 0.52 | 0.69 |
| 5 | 0.75 | 0.35 | 0.69 |
| 6 | 0.77 | 0.36 | 0.69 |
| 7 | 0.77 | 0.35 | 0.69 |
| 8 | 0.77 | 0.35 | 0.69 |

In recent years, the development of water quality index has become popular among government and related agencies for a quantitative measure of water quality status and for the evaluation of river systems as a river water quality management problem [76]. In a typical water quality index, various important water quality indicators will be integrated into a single water quality index, which can be easily communicated among the stakeholders [77]. However, such indexing methods with respect to water quality evaluation system are burdened with uncertainties originating from errors in measurement, imprecision in characterization, classification and weighting system [31]. In this context, few studies have considered fuzzy-based classifications in the evaluation of water quality indices to address the uncertainty in the quality evaluation [78]. Singh et al. [31] considered the attributes of the water quality parameters as linguistic variable and water quality index of a given location was estimated by aggregating the attributes based on degree of importance to develop fuzzy comprehensive water quality index. The study was implemented in various locations on the Yamuna river, India and tried to address the uncertainty due to natural stream flows originating from rainfall uncertainty and corresponding uncertainty in the prediction of water quality by considering the quality attributes as fuzzy variables. In another recent study by Chanapathi and Thatikonda [78], a fuzzy-based inference system was developed for defining the regional water quality index, the fuzzy-based regional water quality index (FRWQI), based on ten water quality parameters to address the uncertainty due to imprecision for the major rivers of India as: Wainganga, Bhima river, Subarnerekha river, Beas river, etc.

## 4. Water Resource Management under Climate Change Induced Uncertainties

Water resource systems management models have been advanced in recent years to consider climate change as a driving force to develop adaptive policies in the decision making [79]. In this context, climate change impact assessment in terms of reservoir operation and altered optimal policies has been widely developed by many researchers all over the world [10,80,81]. The most sophisticated and advanced techniques for the climate change impact assessment studies are statistical downscaling models using the most credible general circulation model (GCM) outputs to predict the projected scenarios of hydrological variables [82]. In this context, a few Indian case studies

made efforts to integrate statistical downscaling models to predict the reservoir inflows under climate change and addressed the associated uncertainty due to various climate model projections. For example, Ghosh and Mujumdar [82] predicted monthly inflows to Hirakud dam, Mahanadi river basin, using fuzzy clustering and the relevance vector machine as a downscaling model. Raje and Mujumdar [83] used the conditional random field (CRF) downscaling model to predict the inflows of Hirakud Reservoir, Mahanadi basin, India. Rehana and Mujumdar [4] used canonical correlation analysis (CCA) to predict the monthly inflows of Bhadra reservoir, India. These studies predicted reservoir inflow projections by considering the influence of various climate variables using statistical downscaling models and GCM outputs. However, these models do not account for the uncertainties of rainfall, catchment characteristics, soil and land use changes in the reservoir inflow prediction. In this context, Shimola and Krishnaveni [84] studied the climate change impact on Periyar reservoir inflows, Vaippar river, by considering a combination of change of precipitation and temperature and regional climate change scenarios by integrating hydrological model, Soil Water Assessment Tool (SWAT) [84]. However, this model does not integrate the modeled reservoir inflow projections along with the reservoir operation model. Such integration can address the uncertainty originating from uncertain climate change projections of reservoir inflows and resulting operating policies. In another study, Adeloye et al. [10] evaluated the hedging-integrated reservoir rule curves on the current and climate-change-perturbed future performance for Pong reservoir, Beas river in Himachal Pradesh, using sequent peak algorithm and genetic algorithm as optimization model.

Climate change impact assessment on river water quality management has also gained much attention in recent years [85,86]. Rehana and Mujumdar [87] employed CCA as a statistical downscaling model with a threshold-based risk of LWQ model based on multiple logistic regression to develop adaptive treatment policies for the projected scenarios under climate change with the Tunga-Bhadra river system as a case study. The model considered uncertainty due to randomness and imprecision in terms of imprecise fuzzy risk with an integration of climate change projection model. The projected decrease in streamflows and increase in water temperatures tend to decrease DO levels and increase the risk of LWQ events along the Tunga-Bhadra river system. The extreme risk of LWQ was predicted to increase by 50.6% for the period of 2020–2040 compared with the current risk levels of 4.5% for the Tunga-Bhadra river system under climate signals [88]. The fractional removal policy may reach up to its maximum limits of 90% during the period 2070–2100, even though the effluents are at safe permissible levels, indicating revised current standards for better river water quality management for future scenarios under climate change uncertainty.

An integrated water resource management model under climate change, as shown in Figure 1, is subjected to a range of uncertainties, including uncertainty due to hydrological models [89], climate model and scenario uncertainty [90] and uncertainty due to downscaling models [91]. Such climate model and scenario uncertainty in the water resource systems can originate due to inadequate information of underlying geophysical processes, the variability of internal parameterization and boundary conditions [92]. Climate change impact assessment studies of water resource management are associated with various uncertainties originating from variation of climate change projections resulting from various climate models, leading to GCM and scenario uncertainty [93]. A few other sources of uncertainties are associated with climate model initial conditions, statistical downscaling models, hydrological models and parameters [94]. In this context, few studies have attempted to address the climate model uncertainties into water resource management [95].

Raje and Mujumdar [96] developed an uncertainty modeling framework for Mahanadi River at Hirakud Reservoir in Orissa, India, to address GCM scenario uncertainties along with uncertainty in the nature of the downscaling relationship with the Dempster–Shafer theory of evidence combination. The results suggest that by linking regional impacts to natural regime frequencies, uncertainty in regional predictions can be realistically quantified. Raje and Mujumdar [3] derived reservoir operating policy for Hirakud reservoir, Mahanadi Basin, India by considering the reliability of hydropower generation for the current scenario, with consideration of conflicts between hydropower, irrigation and

flood control with the standard operating policy (SOP). The projected monsoon streamflows for current and future scenarios for a range of GCM-scenario combinations were used with an integration of a conditional random field (CRF)-downscaling model as the statistical downscaling model to address the uncertainty in the climate model projections. The results of the study found a decrease in hydropower and increase in vulnerability for the future, with a significant impact in terms of a decrease in reliability and increase in vulnerability. The study also suggested revising the reservoir rules under climate change with a projected decrease in inflow to the Hirakud reservoir.

Rehana and Mujumdar [4] developed an integrated regional water resource management model addressing various sources of uncertainties in the prediction of a hydro-climatic variable projection model, an irrigation demand quantification model, and a water quantity quality management model using SDP and fuzzy optimization for Bhadra Reservoir-River system in Karnataka, India. A SDP model is used to derive the optimal monthly steady state operating policy considering irrigation, hydropower and downstream river water quality as reservoir users. The uncertainty due to the randomness of reservoir inflows was addressed using SDP, and the imprecise goals of each reservoir user were addressed by considering fuzzy memberships. A fuzzy water allocation model was developed for obtaining the optimal allocations among various users of the reservoir under climate change.

Another prominent source of uncertainty is the variability of climate projections resulting from different climate models and scenarios, which has been identified as climate model uncertainty in water resource management modelling [93]. Certainly, a range of climate change projections resulting from various models will provide flexibility in decision making [97]. However, combining projections resulting from various GCMs and scenarios to have a single representative projection by deriving a multimodal weighted mean has been widely applied to address climate model uncertainty in water resource management [4] (Figure 4). In this context, Mujumdar and Ghosh [93] proposed a possibilistic approach to address climate model uncertainty, with Hirakud dam inflow climate change projections, located on Mahanadi river, Orissa, India, as a case study. The study developed the possibilistic mean cumulative distribution function (CDF) by assigning weights to GCMs and scenarios based on their performance in the recent years as well as for the future scenarios. The results of the study reveal that the amount of uncertainty for a given inflow projection will increase with time, due to different climate sensitivity among the models. Instead of using a single climate projection resulting from one GCM and scenario, the use of such multimodal ensembles may be promising in water resource management models under climate change.

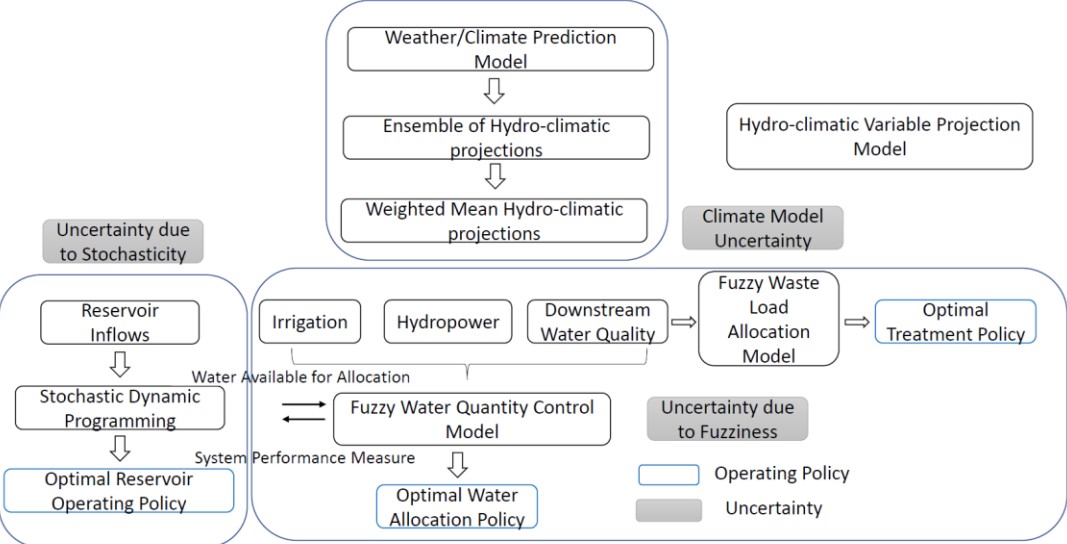

**Figure 4.** Basin-scale water resource systems modeling and associated uncertainties.

In another study by Rehana and Mujumdar [98], entropy weights to each GCM and scenario projections were assigned based on the performance of the GCM and scenario in reproducing the present climatology and deviation of each projections from the projected ensemble average. Entropy weights were assigned to each hydro-meteorological variable defining water availability (reservoir inflows) and demands (e.g., irrigation demands: rainfall and other meteorological variables affecting evapotranspiration, etc.) in the reservoir operation. The multimodal weighted mean (MWM) projections of various hydro-meteorological variables addressing the climate model uncertainty have been used in the water resource management model developed for Bhadra reservoir, India as case study (Figure 4).

Uncertainties are expected to occur at every stage of the water resource management models and their propagation at regional and local scales can lead to large uncertainty ranges and increasing the complexity in decision making [27]. The climate model uncertainty originating from the mismatch between various GCMs and scenarios can be considered as the first level of uncertainty, which can be modeled by using the weighted mean hydro-meteorological projections in reservoir inflow modelling (Figure 5a), and the estimation of projected demands (Figure 5b) in the reservoir operation. The second level of uncertainty originates due to the imprecision and conflicting goals of the reservoir users leading to uncertainty due to imprecision, which can be modeled by using fuzzy set theory. The third level of uncertainty can arise from the inherent variability of the reservoir inflow leading to uncertainty due to randomness, which can be modeled by considering the reservoir inflow as stochastic variable in SDP and consequent uncertainties in resulting operating policies (Figure 5c). Since uncertainties accumulate from various levels, their propagation up to the regional or local level leads to large uncertainty ranges at such scales [27].

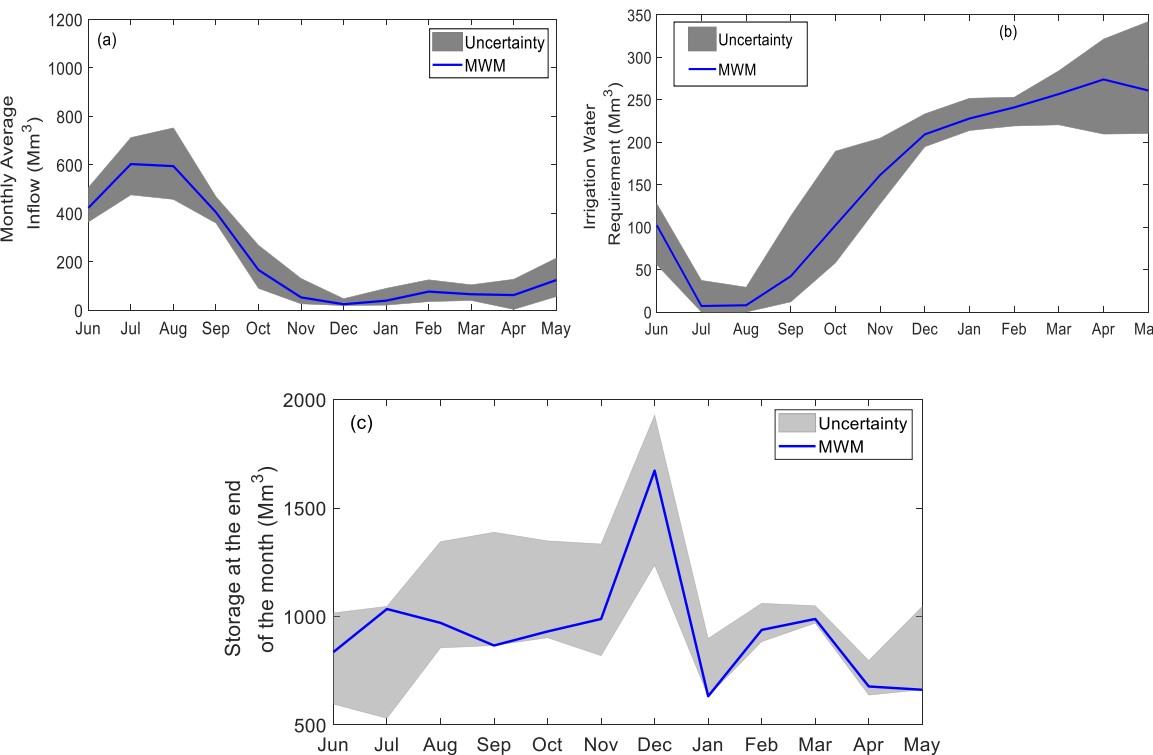

**Figure 5.** Regional water resource management model-associated uncertainties at each stage: (**a**) water availability model in terms of reservoir inflows; (**b**) demand estimation model such as irrigation water requirements; (**c**) operating policies in terms of storages. The results are for Bhadra river basin, India, showing maximum and minimum values along with multi weighted mean (MWM) hydro-climatology.

Overall, while most of the studies tried to address climate model uncertainties in reservoir operation models, the scope of improvements may be towards addressing hydrological model

uncertainties, different downscaling models and various reanalysis data sources of uncertainties, along with the stochastic and fuzziness uncertainties. The assessment of uncertainties in decision making at each stage of reservoir operation has to be understood for the possible risk of failure of water resource systems. This necessitates the development of holistic approaches to include various sources of uncertainties at each stage of the water resource management model, from climate or weather predictions to operating policies.

## 5. Challenges, Missing Links and Directions for Future

Water resource systems management models have emerged as promising tools for the effective management of resources in economically efficient manner in recent years addressing various forms of uncertainties. However, constraints and challenges still remain in terms of the inadequacy of water to meet demands, rapidly growing population, urbanization, increased social and economic development and uncertain future climate [99].

Most of the studies of water resource systems in India have focused on a single reservoir with single or multi objective functions. Integrated water resource management can be developed, considering various reservoirs of a river system, accounting for inflows, uncontrolled intermediate catchment flows, irrigation return flows and controlled flows from an upstream reservoir [12]. The development of an integrated river basin management which can include various reservoirs inter-connected in a river basin and considering various forms of uncertainties can be potential area of research. Such integrated water resource management studies may have challenges in implementation in terms of ungauged sub basins of upstream catchments, a lack of detailed data on downstream releases, lack of information about the inauthentic water abstractions, etc.

A water resource systems model has to integrate various sources of information, such as hydrological, meteorological, pollutant, agricultural, demographic and socioeconomic. One of the major challenges in water resource management models is the integration of various sources of uncertain information which are burdened with spatial and temporal mismatches among scales. Synthesizing various sources of information needs careful attention in terms of validation with field observations not only at the individual scale but also at the integrated scale. Such holistic approaches have the capability to capture the association between various subsystems of regional water resources. In addition, integrated water resource management can improve agreement and cooperation between various stakeholders for sustainable water management.

One of the major challenges which arises in the implementation of holistic approaches at various spatial and temporal scales is the expected increase in climate extremes such as floods and droughts, as the developed water resource systems models are based on the observed and historical data and therefore bounded with the experiences faced in the past and do not consider the possible anthropogenic and climate extremes. The sophisticated climate change impact assessment models developed in recent years can provide a basis to understand projected changes in terms of hydrologic variability and possible adaptive policies. However, such climate change impact assessment studies are developed based on past historical observations under nonstationary assumptions with uncertain information. Future advancements have to be made towards the development of universal water resource management models under hydroclimate extremes along with operation and management by addressing various sources of uncertainties.

The existing water resource management models so far developed are able to address various forms of uncertainties such as randomness, imprecision, fuzziness, inexactness, lack of knowledge and missing data, climate model uncertainties, models and parameters, etc. Most of these uncertainties have been addressed at the individual scale but not in an integrated manner. There is a necessity to integrate various sources of uncertainties to study resulting combined uncertainty and impact on operating policies. Such uncertainty accumulation studies can be promising in the development of approaches representing uncertainties originating at each stage of decision making.

Another area for water resource management model evaluation is towards the development of decision support systems (DSS) for real time operating policies which can act as a bridge to link the model-generated decisions with practical water utility. Such models should work as real time holistic approaches with an integration of weather forecasting models, hydrological models, reservoir operation models and operating policies. The current real time water resource models are dedicated to a single purpose, majorly as a flood controlling devices, with quantity control as a priority. The land use and land cover changes of a natural landscape can intensify the sediments, nutrients and other organic pollutants entering into inland water bodies such as reservoirs, lakes etc. Furthermore, increasing pressure of crop yields has increased the use of fertilizers, which again has increased the number of non-point sources of pollution to the rivers. Under these consequences, real-time water management operating policies should work with an integration of quantity as well as quality as priorities. The development of a general approach which can integrate quantity and quality aspects integrating reservoir-river systems with a DSS in a web-based environment can be a promising tool for the development of real-time operating policies. Formulating such real-time holistic approaches necessitates close coordination and cooperation between various stakeholders, researchers, government bodies and policy makers. It is important to identify all the beneficial and adverse ecological, economic, environmental, and social effects in the context of long-term effects associated in water resource system planning and management.

## 6. Conclusions

Water resource systems models have advanced in several directions, starting with modelling approaches, uncertainty quantification, ease in decision making of stakeholders, along with climate change impact adaptation. Uncertainty quantification in water resource systems models has been identified as an active research topic in the research community. The major source of uncertainty identified in reservoir operation is the random nature of streamflows and this has been addressed using various forms of stochastic dynamic programming. Another major source of uncertainty considered in the research community is the imprecision and vagueness in defining the goals of stakeholders. Such uncertainty was defined as uncertainty due to fuzziness, which has been addressed by considering the goals as fuzzy membership functions and associated satisfaction levels. Fuzzy optimization has been used as a revolutionary algorithm in water resource management models that deals with the uncertainty arising due to fuzzy goals of decision-makers. Fuzzy optimization models in water resource systems have progressed further to address the next level of uncertainty associated with defining the membership parameters by considering them as interval grey numbers. Uncertainty due to a lack of knowledge and missing data has also been tackled by considering the grey fuzzy optimization models. The consideration of hydrological variables as interval grey numbers has resulted in a range of operating policies and provided flexibility to the stakeholders. Climate change-induced uncertainty has emerged as a major source of uncertainty in water resource management models in recent years under changes of hydrological extremes.

To summarize, we reviewed water resource management models and associated uncertainties originating in modelling and decision making. Water resource management models such as reservoir operation, water quantity allocation, waste load allocation, quantity-quality integrated models and water quality index models were discussed. The recent developments in water resource management under climate change were articulated. Several methods that deal with different sources of uncertainties originating in the water resources modelling and decision making were critically evaluated with a focus on Indian case studies. The research gaps, challenges, missing links and future directions in water resource management models under uncertainties were discussed. This review suggests that water resource management models are powerful computational tools that ought to be upgraded by synthesizing various sources of uncertainties for real-time operation and sustainable policy making.

**Author Contributions:** S.R., C.R.R., S.K., S.G., and P.M. conceptualized and designed the paper; S.R. and C.R.R. wrote the paper; P.M., S.K., and S.G. supervised the paper. All authors have read and agreed to the published version of the manuscript.

**Funding:** This research received no external funding.

**Acknowledgments:** We sincerely thank the editor and the three anonymous reviewers for reviewing the manuscript and offering critical comments to improve the manuscript. C.R.R. is funded by the Pacific Institute for the Mathematical Sciences and the Global Water Futures, Center for Hydrology, University of Saskatchewan, Canada.

**Conflicts of Interest:** The authors declare no conflict of interest.

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
