# Peer review of "Uncertainty Quantification in Water Resource Systems Modeling: Case Studies from India"

_water, doi:10.3390/w12061793_

Round 1
Reviewer 1 Report
The paper provides a comprehensive review of water resource systems models accounting for multi-level uncertainties, with a geographical focus on Indian case studies.
The paper can be recommended for publishing in Water journal, provided that the following minor revisions are made:
[Line 19]: ‘In systems modelling’ -> ‘In water resource systems modelling’, or similar? (The phrase at lines 19-23 refers to uncertainties specific for water sector.)
[Line 21]: What is the difference between ‘imprecision’ and ‘inexactness’? Please consider providing a brief comment on this in ‘Introduction’ (or , alternatively, in Section 5, where these terms appear again at line 548).
[Lines 66-69]: Apparently, the two phrases at these lines should be merged in one phrase.
[Figure 1]: Legend improvement suggestion: ‘Water Resources Management Model’ -> ‘Components of Water Resources Management Model’, or similar?
[Line 90]: ‘related’ -> ‘related to’
[Line 91]: Comma after [24] necessary.
[Line 91]: A space near the reference bracket is missing and should be inserted.
The same comment also applies to Lines 93, 97, 101, 105, 156, 161, 184, 303, 430, 432, 470, 521.
[Line 96]: ‘stake holders’ -> ‘stakeholders’
The same comment also applies to Lines 192-193, 230, 277, 366.
[Line 144]: Is an aggregation period of the optimization model necessarily equal to one year? If yes, please provide a brief comment, why it is so.
[Line 160]: Are the authors aware of any models using multistep Markov processes? Please provide a brief comment on this.
[Figure 3]: Adding a reference to the original publication in the figure caption is recommended.
[Line 190]: ‘needs’ -> ‘need’
[Lines 201-203]: Please explain the meaning of function f(.) in Eqs. (5)-(7).
The same comment also applies to Lines 283-284 (Eqs. (11)-(12)).
[Line 209]: Indent is unnecessary.
The same comment also applies to Lines 307, 313.
[Line 232]: ‘address’ -> ‘addresses’
[Line 258]: ‘model’ -> ‘models’
[Line 292]: Please insert a space before ‘and’
[Line 298]: ‘decision maker.’ -> ‘decision makers.’
[Line 305]: Has an abbreviation LWQ been explained above?
[Line 308]: ‘)’ after a formula is necessary
[Line 313]: Please insert a space before ‘of’
[Line 324]: full stop - > comma
[Line 341]: Please provide a brief explanation of the term ‘fat solutions’.
[Line 433]: Comma after ‘GCM’ unnecessary.
[Line 453]: ‘was’ -> ‘were’
[Line 509]: Comma after ‘necessitates’ unnecessary.
[Line 515]: Please delete ‘are’
In reviewer’s opinion, the paper would benefit from providing a table with all abbreviations and acronyms explained (if this would be in line with the journal technical requirements to manuscripts).
Additionally, as the paper tackles the climate change problem, providing a reference to the IPCC Fifth Assessment Report (URL: https://www.ipcc.ch/assessment-report/ar5/) is strongly recommended.
Author Response
"Please see the attachment

Reviewer 2 Report
The Paper is a review of publications on the subject of water management models and associated uncertainties in water resources modeling. The authors aptly describe model components as a combination of water quantity and quality estimation model, water demand estimation model along with the decision-making model. The authors draw attention to the comprehensive management of the catchment area, taking into account the interests of different users and trying to indicate the comprehensive approach to model creation.
The work was prepared on the basis of 92 publications illustrating the development of discussed research issues. The article focuses on examples taken from the area of India. It consists of four main parts relating to reservoir operation, water quality management, the impact of climate change on water management, research directions for the future. This structure is coherent and creates a logical whole. The value of the work is to discuss the main values of individual methods and their development with reference to selected examples, which makes visible didactic values of the work. The authors have characterized the types of models and their development, giving examples of areas where these models have found application. The interested reader will easily find the main features of models and the problems accompanying them and then will easily reach for the source materials. In this context, the publication is a kind of compass. The authors rightly anticipate future challenges in the assessment of uncertainty, paying particular attention to climate change, including sudden weather phenomena. They rightly point out the challenges associated with the development of decisions support system functioning in real-time.
Due to the review nature of the work, it is difficult to indicate critical remarks. I allow myself to point out minor comments and editorial suggestions:
- line 304 - after expressing "low water quality", please add "(LWQ)", which appears afterward,
- line 308 - no parenthesis in the expression specifying probability,
- 340-341 please check the wording "and fat solutions",
- 368-369 citation "Singh et al., 2019" please write as "[30]",
- line 394 after Canonical Correlation Analysis, please add "(CCA)" , which appears on line 409,
- lines 634-635 - please correct the order: author - title,
- line 731 - there is no information about Chapra, S.C. Surface water-quality modeling,
- line 732 - incomplete information on the publication of Chapra, S.; Pelletier, G. Documentation and Users Manual. 2003, 121,
- in figure 3 please describe the legend (flow directions, reservoirs),
- line 221 - please consider specifying the definition of the "q" parameter giving its physical meaning (it is only mentioned that this parameter corresponds to optimum water allocation),
- lines 495-502 - in the description of Figure 5, the titles of smaller figures should be separated by a comma; the descriptions of charts in the word "uncertainty" lack the letter "n",
- references are dominated by letter abbreviations of authors' names, but sometimes full names of authors are given (eg. line 737).
Author Response
"Please see the attachment

Reviewer 3 Report
The authors did a good job in reviewing water resources management models and associated uncertainties. This article recommends water resources management models be upgraded by synthesizing various sources of uncertainties. Overall, this is a solid review article. I don't have any major concerns. Here are some minor comments/suggestions:
- I'm not sure about the role Figure 3 plays in this article. It plots an example of reservoir operations, but how does this translate to the uncertainty estimation? Or more specifically, can you demonstrate the relationship between this operation chart and uncertainty in this figure?
- Line 281-287. the alignment of these equations seem to be off. Can you do some format check here?
- I think it would be an interesting topic to discuss about floods. How would your conclusion change if we consider flood modeling coupled with different water resources modeling?
Author Response
"Please see the attachment
